# The Gut Microbiota and Inflammation: An Overview

**DOI:** 10.3390/ijerph17207618

**Published:** 2020-10-19

**Authors:** Zahraa Al Bander, Marloes Dekker Nitert, Aya Mousa, Negar Naderpoor

**Affiliations:** 1Monash Centre for Health Research and Implementation, School of Public Health and Preventive Medicine, Monash University, Melbourne 3168, Australia; aya.mousa@monash.edu; 2School of Chemistry and Molecular Biosciences, The University of Queensland, Brisbane 4072, Australia; m.dekker@uq.edu.au

**Keywords:** gut microbiota, microbiome, inflammation, cytokines

## Abstract

The gut microbiota encompasses a diverse community of bacteria that carry out various functions influencing the overall health of the host. These comprise nutrient metabolism, immune system regulation and natural defence against infection. The presence of certain bacteria is associated with inflammatory molecules that may bring about inflammation in various body tissues. Inflammation underlies many chronic multisystem conditions including obesity, atherosclerosis, type 2 diabetes mellitus and inflammatory bowel disease. Inflammation may be triggered by structural components of the bacteria which can result in a cascade of inflammatory pathways involving interleukins and other cytokines. Similarly, by-products of metabolic processes in bacteria, including some short-chain fatty acids, can play a role in inhibiting inflammatory processes. In this review, we aimed to provide an overview of the relationship between the gut microbiota and inflammatory molecules and to highlight relevant knowledge gaps in this field. Based on the current literature, it appears that as the gut microbiota composition differs between individuals and is contingent on a variety of factors like diet and genetics, some individuals may possess bacteria associated with pro-inflammatory effects whilst others may harbour those with anti-inflammatory effects. Recent technological advancements have allowed for better methods of characterising the gut microbiota. Further research to continually improve our understanding of the inflammatory pathways that interact with bacteria may elucidate reasons behind varying presentations of the same disease and varied responses to the same treatment in different individuals. Furthermore, it can inform clinical practice as anti-inflammatory microbes can be employed in probiotic therapies or used to identify suitable prebiotic therapies.

## 1. Introduction

The understanding of the symbiotic relationship between the human gut bacteria and the overall functioning of the body has significantly deepened and broadened, with acceleration in research output concerning this area. While initial research generally examined the microbiota composition and its relation to disease presentation, there has recently been a more prominent shift towards the understanding of the mechanisms by which variation of the microbiota can lead to disease manifestation [1]. Clarifying these mechanisms can inform the development of novel therapies and optimise clinical practice. Moreover, discerning the structure and function of the microbiota becomes increasingly important in the era of precision medicine as this allows for personalised treatment to improve clinical outcomes [2]. This review presents a brief introduction to the gut microbiota, influential factors on microbiota, analysis techniques of the microbiome and interpretation of results. These are followed by a review of the recent literature on the interplay between the gut microbiota, metabolic diseases and inflammation. We explore potential underlying pathophysiologies and the intermediate biomarkers associated with gut bacteria and inflammatory diseases. Finally, limitations of microbiome studies are discussed, with recommendations for future research directions.

## 2. What Is the Gut Microbiota?

The gut microbiota refers to the microorganisms inhabiting the human gastro-intestinal tract. This dynamic community predominately encompasses species from the prokaryotic domain (comprising bacteria and other unicellular microbes lacking specialised organelles) and, to a lesser extent, fungi, parasites and archaea. Viruses are also constituents of this environment. The genetic and functional profile of microbial species is termed the gut microbiome [3]. On a broader level, humans can be classified according to their enterotypes, which are distinctly similar gut microbial compositions that can be observed in certain populations [4]. In a single individual, the inhabiting gut microbial species collectively have been shown to contain 3.3 million genes which, when compared to the human genome’s ~23,000 genes, demonstrates the sheer magnitude and potential effect of these species on human health [5]. Moreover, the human body contains almost as many bacterial cells as human cells [6].

Bacteria can be classified through phylogenetic nomenclature into various levels of groups termed taxonomic ranks, dependent on genetic similarity. Organisms that are within the same lower taxonomic rank groups have greater genetic sequence similarities, indicative of more recent evolutionary divergence [7]. The modern classification system is illustrated in Figure 1. Members of the same species are most similar.

## 3. Methods for Studying the Gut Microbiota

Researchers study the gut microbiota through various approaches, and it is the combination of these different methods that enables a comprehensive construction of knowledge revolving around gut microbiota and health. In human studies, the most established method of characterising the gut microbiome is stool sampling as it is readily available, non-invasive and densely populated by microbes representative of the luminal intestinal gut microbiota [9,10].

Following collection, faecal samples are frozen and stored immediately, generally at −80 °C [11]. DNA is extracted from the faeces with two major steps. First, the sample is purified through the use of multiple reagents and centrifugation, allowing for the microbes to be stripped of other components of the faeces. The subsequent step involves lysing bacterial cells by incubating the samples with lysis buffer with agitation, such as vigorous shaking with or without beads, and carrying out further centrifugation [12]. Following this, the resulting DNA is amplified using approaches such as multiple displacement amplification [11]. A 16S rRNA primer is selected and used for gene sequencing. The sequence data obtained undergo filtering to ensure the thresholds of quality are met. Subsequently, the sequence count is normalised prior to operational taxonomic units (OTU) analysis; the method by which related bacteria are grouped together. An OTU clustering algorithm is applied to identify bacterial genera and species [11]. The general protocol is summarised in Figure 2.

Microbial diversity is assessed according to two parameters; alpha and beta diversity. Alpha diversity refers to the variation within a single sample, whilst beta diversity refers to that between samples [13]. These are generally determined in increasing order of relatedness, from phylum to genus, although some studies also employ species-level analysis [14]. Platforms facilitating these analyses include QIIME, a free software tool that evaluates 16S rRNA sequences [13]. However, using this lower rank classification could yield unreliable findings; most studies typically only assess one portion of the 16S rRNA gene as opposed to examining the full-length of the gene [15]. Doing so compromises the reliability of species-level classification as a small portion of the gene can have a highly variable ability to detect species [15]. As reagents used in these steps may introduce contaminants, it is important to test for contamination with 16S rRNA gene amplification, and computationally erase the contaminant species’ sequences from the overall sequences obtained [11].

A more representative snapshot of the gut microbiota can be obtained through mucosal tissue sampling of the distal gastro-intestinal tract; however, this method comes with various limitations and is, therefore, less commonly used in research [16]. One such limitation is that compared to stool sampling, where human cell contamination is minimal, analysis of mucosal tissue biopsy samples must take into consideration the presence of human cells and genetic material as these may interfere with findings [10]. Other methods less routinely used include lavage and swab sampling, however, these have not been extensively evaluated in terms of protocol consistency and how representative the findings are of the gut microbial community [17].

Studies utilising animal models are key and frequent in this area of research as they enable a more comprehensive understanding of some aspects of the role of the gut microbiota in health. Confounders can be controlled more strongly in animal models, including rodents, fish and insects, to better understand experimental variables and their effects, and host–microbiota interactions [10,18]. This is particularly relevant in studies that seek to examine factors that affect the gut microbiota, or the effects of gut microbes on tissues or systems. Germ-free animal models have been utilised to assess microbiota-independent functioning in comparison to normal animal models [19]. Furthermore, gnotobiotic models are used in this field of study, where animals, typically rodents, are prepared with specific known strains or combinations of bacteria and are often genetically altered to determine the downstream effects of certain bacterial products or associated products [20,21,22,23,24]. Nevertheless, the results of studies on gnotobiotic models need to be interpreted with caution since the immune system of these animals is altered from early in development.

Cell culture systems are also important here as they can be used to further examine controlled interactions between gut microbes and other variables. These techniques may involve the cultivation of native-to-human-gut microbes, human tissue, or artificial tissue [10].

## 4. Factors Associated with Microbiota Composition

The gut microbiota in the gastro-intestinal (GI) tract develops during infancy. The process of microbial colonisation is contingent on a variety of host factors, described in more detail below. Additionally, different regions of the GI tract have different chemical environments that allow certain microbes to thrive more than others. Diversity and stability continually increase throughout early development, reaching a certain composition with age [25]. This composition may be modulated by various factors and life events, that may synergise to bring about greater diversity [1]. It is important to note that here, greater diversity is generally associated with better health outcomes, although there have been some reports of increased diversity in certain disease states [26,27]. Factors that influence the gut environment will in turn contribute to the characteristic gut microbial ecosystem thus giving rise to the inter-individual variation we observe.

### 4.1. Mode of Delivery and Feeding

Species that partake in the initial colonisation of the GI tract differ depending on the mode by which a neonate is born [28]. Both caesarean section and vaginal modes involve exposures to microbes, however, differences come about regarding the source of these microbes. In vaginal delivery births, the neonate is exposed to maternal vaginal bacteria and therefore, their initial microbiota compositions tend to reflect this region [29]. More specifically, these bacteria are part of the *Lactobacillus*, *Prevotella* and *Sneathia* genera. In contrast, those delivered by caesarean section are strongly influenced by their maternal skin flora and tend to have less diverse microbiotas [29,30]. As such, they have a higher portion of skin dwelling microbes including *Staphylococcus*, *Corynebacterium* and *Propionibacterium* species [29]. Additionally, intrapartum antibiotic use during casesarian section is shown to affect the neonatal gut microbiota [31]. Overall, a greater degree of similarity in gut microbiota has been observed between newborns delivered vaginally and their mothers, compared to babies delivered by C-section [32]. The role of the microbiota is critical in the development of the immune system, therefore differences in gut microbiota may explain why the latter category have a greater risk of developing infections or allergies [33,34].

In addition to the mode of delivery, the type of milk consumed early in life also affects the gut microbiota. Breast milk differs from formula milk as it contains its own bacteria and different bioactive compounds and nutrients. Specific species of bacteria may be better adapted for guts rich with their preferential nutrient, for example, dietary fibre, as they are able to extract their energy from these [35]. Because of this, *Bifidobacterium* species predominate in the microbiota of breastfed individuals, whereas *Enterobacteriaceae* species are more frequent in formula-fed infants [30]. Further, formula-fed babies have more diverse microbiota than their breastfed counterparts, suggesting that the benefits of breast-milk may be due to factors unrelated to the gut microbiota [32]. It is important to note that the gut microbial community undergoes many changes during the initial stage of development of a newborn, and that the effects of intrinsic and extrinsic factors may only lead to temporary and ephemeral structural changes [36].

### 4.2. Diet

Dietary behaviour can result in some strains dominating the gut more than others, and therefore specific downstream phenotypic characteristics in the host. Modifications to the diet will make different nutrients more readily available, consequently changing which strains dominate, contributing to the dynamic nature of the microbiota [37]. It has been revealed that individuals consuming more meat in their diets have significantly different gut microbiomes to those with plant dominant diets, due to certain bacteria flourishing in the abundance of protein [38,39]. For instance, *Bacteroides* species tend to dominate in the GI tract of those consuming animal protein, and *Prevotella* species are associated with plant-based diets [40]. Moreover, the effect of diet on gut microbiota can be seen in ethnography studies, where culture affects the general kinds of food consumed, and hence collective enterotypes [41]. This can be observed within Asian populations that have a high consumption of starch-rich foods like rice; the enterotypes of these populations are characterised by a notably high abundance of *Bifidobacterium*, a genus known to produce large quantities of starch-metabolising enzymes [4]. The Western diet, rich in saturated fats but low in unsaturated fats, has been studied extensively and found to be positively associated with anaerobic microorganisms and specific genera including *Bacteroides* and *Bilophila* [42]. In addition to the effects of specific macro and micronutrients in human diets on the gut microbiota, some animal studies have noted that additives commonly found in the Western diet have associations with altered microbiota composition and its inflammatory potential [43].

### 4.3. Age, Sex and Body Mass Index

Differences in the composition of microbiota have been noted with respect to age. In a cross-sectional study of 367 healthy participants in Japan, similar phyla were observed in the gut microbiome across age groups ranging from 0–104 years, although in different proportions [44]. One trend reported was an association between age and abundance of Actinobacteria, where an increase in age was associated with a decrease in abundance. Microorganisms of the phylum Firmicutes became abundant in the microbiotas of individuals post-weaning stage, particularly past the age of 4, whilst the inverse was seen with the Actinobacteria population. Bacteroidetes maintained a stable level of abundance across different age groups, although individuals above the age of 70 had higher levels of these microbes [44]. The patterns found in this study may be attributed to the various events that occur over the life course, including varied environmental exposures to microorganisms, occurrence of disease, hormonal changes and immune system functioning, thus complicating the influence of age on the gut microbiota [44,45].

Although studied far less than other factors, sex differences have been noted to influence microbiota composition [46]. Differences become more apparent when investigating the microbes on a genus or species level. In an experimental study by Baars et al. [47], sex-related differences were found in how the gut microbiota interacts with the host. Using mice, they determined that compositional and transcriptional dissimilarities were apparent. In turn, these influenced the differences in lipid metabolism observed between male and female mice.

Haro et al. [48] found that *Bacteroides caccae* were more abundant in females than in males, whilst the inverse was observed for the species *Bacterioides plebeius*. Interestingly, these sex differences interrelate with body mass index (BMI). In the same study by Haro et al. [48], sex variation became more prominent when BMI increased. For instance, males exceeding a BMI of 33 had lower Firmicutes abundance than those of lower BMI, however, this trend was not observed in females. Rather, Firmicutes abundance remained high irrespective of BMI in female participants [48]. These observations are hypothesised to be due to hormonal differences, however, more research must be conducted to understand the exact mechanisms [49]. In general, an association with BMI and gut bacterial diversity has been reported, with a pattern demonstrating that an increase in BMI strongly correlates with decreased diversity [50]. Further, there is a reduced proportion of members of the phylum Bacteroidetes compared to Firmicutes in obesity, and upon weight loss, this ratio reverts to normal [51].

### 4.4. Host Genotype

Specific host genotypes are associated with differences in gut microbial diversity, as well as specific community structures [52]. As genes affect immune system functioning and susceptibility to disease, certain alleles are associated with certain compositions. For example, individuals with the rs651821 variant of the gene *APOA5* are more likely to have members of the *Lactobacillus*, *Sutterella* and *Methanobrevibacter* genera in their microbiota, which subsequently correlates with a greater risk of metabolic disorders [52]. Specific variants of microbiome-associated genes have been shown to be associated with conditions including obesity, shizophrenia, type 2 diabetes, amyotrophic lateral sclerosis and inflammatory bowel disease, in a number of studies [53]. One such example is the SLIT3 gene; this gene appears to play a role in microbe product-induced inflammation, and in a genome-wide association study, variants of the gene were found associated with BMI, suggesting a likely relationship between the gene, its molecular product, the human microbiome and obesity [53].

One polymorphic aspect of human genetics associated with predisposition to autoimmune disorders is the human leukocyte antigen (HLA), which are cell-surface proteins that are encoded by a cluster of genes. The variation in HLA structure and underlying genetics has been shown to correlate with distinct changes in the gut microbiota [54]. For instance, the relationship between the variant HLA-B27 and the inflammatory condition ankylosing spondylitis has been well documented in the literature [55]. One study examining individuals with this particular genotype found that there is a distinct gut microbial profile associated with HLA-B27, suggesting that ankylosing spondylitis may be driven in part by a genetic predisposition to specific immunological processes, mediated by the gut microbiota [56].

It should also be noted that existing findings lend to the idea that the composition of the human gut microbiota is influenced mainly by non-genetic factors [57]. Nonetheless, this area of study is still in its infancy and further studies in larger cohorts are underway to better understand how the gut microbiota and human genome interrelate.

### 4.5. Antibiotic Use

When exposed to antibiotics, the host microbiota can undergo quick alterations in its structure, depending on the type and regularity of use. A common outcome is a state known as gut microbiota dysbiosis, whereby there is an imbalance within the microflora, consequently impairing its functioning [58]. Other factors, including diet, infection and existing disorders, may also cause dysbiosis. An individual with dysbiosis will face an increased risk of a plethora of morbidities, namely infection, as a healthy, balanced microbiota serves as a hindrance to potential pathogenic strains, competing for resources and preventing the growth of invading microbes [59]. Other outcomes associated with dysbiosis pertain to immune system and metabolic deregulation. Findings from a large international cross-sectional study of 74,946 individuals exemplify the effects of antibiotic-induced dysbiosis; it was found that the use of antibiotics within the first year of life in male subjects contributed to metabolic dysregulation and therefore an increase in BMI in later stages of childhood, but not in females [60].

## 5. Role of the Gut Microbiota in Immunity and Inflammation

Microbes possess a variety of functions that influence their ability to grow and colonise, whilst bringing about downstream effects for the host that may be beneficial or otherwise [61]. Humans are not capable of digesting some components of dietary fibre due to the lack of the required enzymes to break down and harness the energy of these carbohydrates [62]. Certain species of microbes produce specific enzymes that enable fermentation of nutrients into absorbable forms, including that of indigestible carbohydrates into short-chain fatty acids (SCFAs) [62,63]. These SCFAs may have anti-inflammatory and immunomodulatory effects [63]. SCFAs are only a small part of the bigger picture as, in addition to enzymes and other metabolites produced, components of the bacteria themselves, including lipopolysaccharides, cell capsule carbohydrates and other endotoxins, may also be released and result in secondary effects to the host. These effects include maintenance of gut epithelium (and thereby integrity of the gut wall), production of vitamins, and interactions with several key immune system signalling molecules and cells, activating and inhibiting specific responses [1]. In addition to nutrient metabolism, gut microorganisms affect aspects of pharmacokinetics as they carry out drug metabolism [64]. They provide a natural defence against pathogenic species through competition and maintenance of the mucosa. It is through their contact with the immune system that the microorganisms occupying the gut can elicit or prevent inflammation. They may be associated with anti-inflammatory mechanisms, stimulating regulatory cells of the immune system to inhibit inflammation [65]. On the other hand, as bacteria regulate the permeability of the intestines, certain species can promote a “leaky gut”, where metabolites associated with the microbes leave the gut and enter the bloodstream. In response, the body produces cytokines and other mediators, effectively launching an inflammatory response [66]. Similarly, cells within the epithelial tissue of the gut deliver bacterial metabolites to immune cells, promoting inflammation on both a local and systemic scale. The persistence of this condition may lead to subacute or chronic inflammation, which may subsequently drive the development of diseases such as inflammatory bowel disease, diabetes or cardiovascular disease [65].

## 6. Proposed Mechanisms and Relationships between the Gut Microbiota and Inflammatory Markers

### 6.1. Lipopolysaccharides

Lipopolysaccharides (LPS), also known as endotoxin, form the key cell wall component of Gram-negative bacteria. Increased levels of LPS are observed in obesity and other metabolic disorders as well as adipose tissue inflammation and pancreatic beta-cell dysfunction [23,67,68,69]. Under normal conditions, the gut barrier including intestinal epithelial and mucosal layers, minimises movement of LPS from the bowels into the systemic circulation [70]. Disruption of this barrier by factors such as diet or pathogenic bacteria, may lead to LPS dislocation and movement between the junctions of the intestinal barrier into the circulation [70]. This leakiness and dysregulated permeability of the gut means macrophages can infiltrate the region, produce and activate inflammatory cytokines, leading to local inflammation [70]. Moreover, LPS binds to toll-like receptor 4 (TLR-4) found on immune cells, and in doing so can activate pro-inflammatory cascades both local to the intestine and in distant sites [67]. In a study by Cani et al. [71], mice fed a high-fat diet were found to have a high amount of LPS circulating in their bloodstreams, exhibiting endotoxaemia, however when administered antibiotics, this elevated level of LPS was not found. This demonstrates that gut microbiota induced by high fat diets release LPS, and when targeted by antibiotics, downstream effects of LPS would be prevented. Here, it is presumed that the antibiotics eliminate most of the microbiota, including LPS-secreting bacteria in locations other than the GI tract.

Upon injecting healthy human participants with LPS, Mehta et al. [72] found increased insulin resistance as insulin receptors in adipose tissue became suppressed, although pancreatic beta-cell function was not compromised. Furthermore, a study by Tian et al. [73] reported that probiotic therapy with *Lactobacillus paracasei* reduced and reversed the effects of LPS-associated pathology in rodents, specifically reducing type 2 diabetes-associated mechanisms such as beta-cell dysfunction. However, it is worth noting that not all members of the *Lactobacillus* genus have anti-diabetic properties, as in a human study it was found that *Lactobacillus gasseri* was elevated in those with type 2 diabetes [74].

### 6.2. Short-Chain Fatty Acids

As stated above, gut bacteria possess the capability of metabolising complex carbohydrates, otherwise undigested by the host, into SCFAs. SCFAs play a critical role in the interplay between diet, the gut microbiota and downstream activation or inhibition of inflammatory cascades. Additionally, they contribute to the homeostatic control of energy and appetite regulation through their effects on metabolic pathways [75]. Depending on the type and concentration of SCFA, their effects on inflammation differ, and SCFA levels may vary in obese and lean phenotypes [76]. In many animal studies, SCFA butyrate has been associated with a variety of roles that oppose the onset of metabolic disorders [40]. Through epigenetic interactions, butyrate promotes lipolysis and mitochondrial functioning in adipocytes, thus enabling a greater energy expenditure and preventing the onset or maintenance of obesity [77,78]. Butyrate is an anti-inflammatory metabolite that is known to inhibit the pathways that lead to the production of pro-inflammatory cytokines [79,80]. In a clinical study of 13 patients with Crohn’s disease, oral administration of butyrate was found to alleviate inflammation in nine patients [81]. Additionally, butyrate minimises the risk of development of insulin resistance as it improves insulin signalling [82]. Butyrate has also been shown to minimise LPS translocation in the intestines, thus reducing LPS-associated effects [40]. *Faecalibacterium prausnitzii* has been identified as a butyrate-producing bacterium with an inverse relationship with different pro-inflammatory markers [83]. In obese individuals, *F. prausnitzii* abundance is reduced compared to individuals with healthy body weight [84].

Acetate, another SCFA, is an important molecule for the processes of lipogenesis and gluconeogenesis. Acetate can serve as a substrate for cholesterol synthesis, thus contributing to elevated serum cholesterol levels [85]. In a study of rats, contrasting with butyrate, acetate was found to correlate with greater insulin resistance and an increase in ghrelin secretion as a result of its activation of the parasympathetic nervous system [86]. As ghrelin is an appetite-stimulating hormone associated with greater food intake, acetate may consequently be linked with weight gain [86].

Although there is some consensus on SCFAs and their metabolic properties, findings have been inconsistent [87]. Regarding the potential role of acetate in obesity, a study conducted by Frost et al. [88] suggested that acetate is associated with appetite inhibition and therefore a lower risk of obesity. However, Perry et al. [86] reported that acetate was linked with an increased risk of obesity. Variation in the research participants, the employed research methods and analytical techniques may account for these observed differences. Nevertheless, there is a clear need for further research to elucidate the exact mechanisms by which SCFAs relate to changes in inflammation, microbiota composition and the subsequent development of metabolic disorders.

### 6.3. Bile Acids

Bile acids have a range of roles in the body, from enabling emulsification of lipids to allow their absorption by the body, to functioning as signalling ligands [67]. The conversion of primary bile acids to secondary bile acids is carried out by gut bacteria. Changes in the gut microbiota influence the types of secondary bile acids that are synthesised, which in turn, may affect the secretion of appetite hormones from the gut, resulting in different appetite inclinations [89].

Of particular relevance to this review is that bile acids, via activating the farnesoid X receptor (FXR) signalling pathway in enterocytes and adipocytes, cause inflammation [90]. Parséus et al. [20] found that mice lacking functional FXRs did not accumulate adipose tissue in the same way as mice with the receptor, suggesting that the gut microbiota may promote the onset of obesity through secondary bile acid formation.

### 6.4. C-Reactive Protein

C-reactive protein (CRP) is an acute-phase reactant associated with cardiovascular disease, type 2 diabetes and obesity [21]. When macrophages and T cells secrete interleukin (IL)-6, this molecule is subsequently released at high levels into the circulation [91]. In a study conducted by Xu and Song [21], the proportion of *Akkermansia muciniphila* declined in obese mice with elevated plasma levels of CRP. However, this does not explain whether CRP modulates the gut microbiota, or whether the gut microorganisms contribute to its elevation.

The abundance of members of genus *Phascolarctobacterium* has been associated with lower levels of CRP [92]. This relationship could potentially clarify why a decline in the proportion of this genus is associated with colonic inflammation [93]: *Phascolarctobacterium* are producers of propionate, an SCFA that inhibits pro-inflammatory cascades by suppressing the activity of pro-inflammatory regulator nuclear factor kappa-B (NFκB) (see Section 6.5) [94,95]. Similarly, the abundance of *Faecalibacterium* is inversely correlated with levels of CRP [81,96]. Thus, CRP is a downstream inflammatory marker that can be down-regulated through the effects of anti-inflammatory metabolic products of specific gut microbes. the evaluation of baseline serum and microbiota data of healthy subjects with BMI over 25 showed that those with higher CRP levels had a significantly lower abundance of bacteria from the genera *Lactobacillus* and *Bifidobacterium*, but a higher abundance of *Escherichia* and *Bacteroides* [97].

### 6.5. Cytokines

Cytokines are signalling molecules secreted by immune cells that affect numerous processes within the body, including immunomodulation [98]. Many transcription factors regulate the production of cytokines, however, the focus of this review will be on NFκB, a regulator deemed prototypical and involved in the activation of various inflammation-related genes [99]. Cytokines are generally classified as either pro-inflammatory or anti-inflammatory in their downstream effects, although this binary classification is simplistic and does not account for the fact that specific cytokines can play opposing roles in multiple processes, and are context-dependent [100]. A detailed description of evidence related to all cytokines and the gut microbiota is beyond the scope of this review. In the following sections, we present some of the established correlations between the gut microbiota and TNF alpha and interleukin-6, the two cytokines that have been more commonly studied.

Table 1 highlights the general tendencies of various cytokines, as reported in previous studies.

#### 6.5.1. Tumour Necrosis Factor-Alpha

Tumour necrosis factor-alpha (TNF-α) is a key cytokine known to drive inflammation. Elevated levels of this molecule have been shown to be associated with insulin resistance and glucose intolerance as TNF-α is able to activate various signalling pathways, including the mTOR pathway, making it a critical molecule in the development of metabolic disorders [104]. Gwozdziewiczová et al. [105] reported that TNF-α was higher in females than males suggesting that there may be sex-related differences in how TNF-α drives disease; however, the study had a small sample size (*n* = 70) subdivided into smaller groups based on sex, and results require confirmation in larger cohorts. In obesity, TNF- α is secreted by adipose tissue macrophages [106] and is linked with different gut microbial species. For example, Schirmer et al. [107] revealed that people with higher *Bifidobacterium adolescentis* abundance had a lower TNF-α production.

#### 6.5.2. Interleukin-6

When adipocytes begin to function aberrantly, usually due to increased macrophage infiltration within adipose tissue, they release the cytokine IL-6 [104]. This molecule drives inflammation by promoting insulin resistance and metabolic dysregulation [108]. Various studies have examined the relationship between IL-6 and type 2 diabetes, finding IL-6 to be predictive of its onset [109,110]. Moreover, IL-6 is positively correlated with the abundance of *Lactobacillus* species [111] but it is unclear whether it is the specific gut microbial composition that results in the elevated IL-6, or the inverse. The abundance of *Faecalibacterium* has been negatively associated with levels of IL-6, a link that may be explained by the genus’ production of butyrate and its consequent inhibition of NFκB [81]. Similarly, a study examining a population of obese individuals showed an inverse relationship between *Faecalibacterium* and IL-6 in both diabetic and non-diabetic individuals [96].

### 6.6. Trimethylamine N-Oxide (TMAO)

Dietary choline, carnitine and betaine, compounds present in red meat, fish and other animal sources, are metabolised by gut microbes into trimethylamine (TMA), which is then converted into TMAO by the actions of host hepatic flavin monooxygenases [112,113]. Examples of TMA-producing bacteria include members of the genera *Clostridium*, *Proteus* and *Escherichia;* however, these may differ with respect to the substrates they utilise to produce TMA [114]. When present in high serum levels, TMAO has been linked with deleterious effects including endothelial dysfunction, which in turn promotes vascular inflammation, atherosclerosis and other cardiovascular disease risk factors [113,115,116,117]. These effects can be explained by the immunomodulatory effects of TMAO which have been reported in various animal and human studies. For instance, Seldin et al. [118] found that increased serum levels of TMAO induced a rise in the production of IL-6 and TNF-α by the NF-κB signalling pathway in mouse and human cell cultures. Moreover, Rohrmann et al. [119] identified a positive association between TMAO levels and TNF-α, but not IL-6 in humans. A study in a healthy population found a positive correlation between levels of TMAO and CRP, however, it should be noted that this study also evaluated the production of TMAO in response to the ingestion of eggs [120]. A similar TMAO and CRP relationship was noted in a different study examining a population of patients with heart failure [121]. Research examining TMAO and inflammation typically utilises a sample with a specific diet (e.g., high protein diet), or with underlying chronic diseases [117,119,120,121]. As such, the role of the gut microbiota independent of diet is difficult to ascertain from the available evidence.

## 7. The Interplay of the Gut Microbiota, Inflammation and Diseases

Constantly heightened levels of inflammatory mediators can initiate pathological processes that may lead to several chronic disorders [65]. As depicted in Figure 3, while the microbiota is proposed to contribute to subacute systemic inflammation, it may also be shaped by the outcome, reinforcing the disease state. As such, it is difficult to determine a single causal pathway. Systemic inflammation can give rise to various conditions including metabolic syndrome, inflammatory bowel disease and cancer, and these have been associated with altered gut microbiota and studied widely in both animal and human studies [20,23,27,71,122,123,124].

### 7.1. Metabolic Syndrome and Associated Disorders

Risk factors including insulin resistance, glucose intolerance, hyperglycaemia, high blood pressure, dyslipidaemia and obesity have all been correlated with variations in microbiota composition [122,125,126,127,128,129]. The collective term for these conditions is metabolic syndrome [130]. This encompasses the cluster of disorders that are precursors to type 2 diabetes mellitus (T2DM) and cardiovascular diseases [130].

Studies in animals and humans have identified circulating levels of LPS as one of the key links between the gut microbiota and inflammation in metabolic syndrome [131,132,133]. Dislocated LPS can stimulate pro-inflammatory pathways through the activation of the receptor TLR4 on adipocytes, and the up-regulation of NF-κB, in turn promoting insulin resistance [23,131]. Individuals with T2DM, obesity and glucose intolerance have higher circulating levels of LPS compared to those unaffected by these conditions [132]. Increased serum LPS levels are also associated with pro-inflammatory cytokines including TNF-α and IL-6 [133]. Circulating pro-inflammatory cytokines may inhibit insulin signalling, promoting insulin resistance [91]. LPS structurally varies among microbes with respect to its ability to activate inflammatory cascades, however notably, members of the Gram-negative phylum Proteobacteria, particularly the family *Enterobacteriaceae* are known to have highly immune-stimulatory LPS [134,135]. In one study, *Enterobacteriaceae* isolated from the gut microbiota of an obese human was used to develop a gnotobiotic mouse model containing only this microbe [136]. Compared to germ-free mice exposed to the same conditions, the gnotobiotic mice demonstrated elevated levels of LPS, and developed insulin resistance and obesity, whilst their germ-free counterparts exhibited normal phenotypes. This demonstrates a link between gut microbes that trigger the production of pro-inflammatory mediators, and the downstream effects of inflammatory and metabolic disorders [136]. In contrast, bacteria known to stimulate anti-inflammatory cytokines such as IL-10 and IL-22, have been associated with anti-diabetic effects in animal models [137]. For instance, the species *Roseburia intestinalis* is positively associated with IL-22, which has been shown to improve insulin sensitivity in mice, and is negatively associated with T2DM [137,138,139].

### 7.2. Cancer

The interplay between the gut microbiota, inflammation and cancer has been noted in several studies. Altered microbiota associated with chronic inflammation has been reported in various types of cancer including pancreas, gastric, colon, liver, breast and prostate cancer [123]. There is strong evidence for gut microbial influence on tumour formation local to the gastro-intestinal tract and on response to cancer treatment [140]. A study examining mice with colon cancer found that tumour development was reduced by the administration of antibiotics, thus indicating a pivotal role for the microbiota [141]. The gut microbiota can influence the development of tumours in numerous ways, including through the production of metabolites that cause genetic instability in host cells [142]. However, for the purpose of this review, only the links between microbiota and inflammation will be explored in the context of cancer. Pro-inflammatory micro-environments are generally associated with amplifying the development of many cancers [143]. This may clarify why long-term IBD is a strong risk factor for the development of colorectal cancer [144,145]. Dysbiosis has been noted in patients with colorectal cancer, where there is a tendency for reduced microbial diversity compared to healthy individuals, however, the taxa of microbes that are overrepresented or underrepresented are variable among cases [146,147,148]. One inflammation-related mechanistic link between microbiota and cancer is the NF-κB pathway, as demonstrated by a study of mice deficient of TLR-4 (a receptor of LPS and an upstream regulator of NF-κB), whereby tumour development was reduced compared to wild type mice [149]. Similarly, the gut microbiota can influence the expression of CCL5 which regulates the signalling of IL-6, enabling the proliferation of epithelial cells that can develop into tumours in colorectal cancer [150]. Further, the composition of the gut microbiota can influence the presence of pro-inflammatory-cytokine-producing immune cells, which in turn can promote a pro-inflammatory environment, considered stimulatory for tumour development [151].

## 8. Limitations in Current Evidence

Despite the increasing number of studies investigating microbiota in relation to inflammation and metabolic diseases, there remain several limitations in the current evidence base. Firstly, most studies utilise faeces to understand the gut microbiome. Faeces contain many impurities associated with the microbes, affecting the quality of the DNA yielded in the extraction phase, thus affecting interpretation of sample alpha diversity [12]. Moreover, as the gut contents change throughout the GI tract, the microbial community present in faeces is more suited to the depleted nutrient environment of the latter regions of the GI tract [152]. As such, faecal microbiota findings should be regarded as snapshots of the gut microbiota, to acknowledge the dynamic nature of this ecological community. Further, differences in stool consistency, compact or loose upon collection, may further influence interpretations of gut microbial activity and therefore must be considered. A human study found that stool consistency is a more important factor in the variation of faecal microbiota, compared to use of medication, early life events and diet [153].

Additionally, there is a need for optimising standard protocol, including faecal collection kits, transportation conditions, storage status, and DNA extraction methods, to allow for better comparisons between studies. This suggestion extends to animal studies as there are variations in methodologies between different laboratories, including diet, stress and circadian rhythms experienced by the animals [154]. For instance, the exact composition of the high-fat diet detailed for one rodent study may differ significantly from another. As different sources of fat result in different digestion outcomes, it is imperative to create a standard for how experimental animals are set up and controlled. Similarly, the environments in which rodents are kept may also play a role in composing the faecal microbiome, as it was observed that certain volatile compounds present in bedding can influence the final microbial readings [154]. Extrapolation of findings revealed in animal studies must be carried out cautiously as there are differences in human and rodent gut microbiomes. Moreover, when germ-free rodents are prepared for study, their immune system development is also altered, therefore findings must be interpreted with caution. Animal studies can broaden our understanding of this field, as they allow for more flexibility and additional options not available in human studies, such as the option of euthanising rodents to assess caecal microbiome samples. While useful, this creates further complications in extrapolating findings, making it difficult to ascertain potential similarities between human and rodent studies in the interplay between microbiota and inflammation.

A very small number of studies have analysed a wide range of inflammatory markers in connection with the gut microbiota. Many have outlined the changes to gut microbial composition with dysbiosis in various conditions, but the mechanisms involved including the levels of cytokines in relation to specific taxa of bacteria within certain phenotypic populations have not been thoroughly examined. In the few studies that assess this link, study design limitations including the presence of comorbidities in participants or the use of medication/s and unadjusted analyses for confounding variables such as diet and physical activity reinforce the need for more studies to confirm these associations.

## 9. Conclusions and Future Directions

Inflammation underlies many of the diseases including T2DM, cardiovascular disease, IBD and cancer which significantly contribute to global morbidity and mortality. A vast pool of studies in animals and humans have indicated a critical interplay between the gut microbiota and inflammation that could inform therapeutic intervention for the treatment of these disorders. Currently, several biomarkers of inflammation are not considered to be good predictors of chronic inflammation [155,156]. More robust biomarkers that depict the onset or degree of progression of disease may be identified through further understanding of the mechanisms bridging the gut microbiota, inflammation and morbidity.

Some microbial species have been associated strongly with anti-inflammatory roles, in particular, *Faecalibacterium prausnitzii*. These microbes can be further investigated in large prospective studies, which in turn may potentially inform trials in prebiotic and probiotic therapy for the amelioration of conditions underscored by inflammation. It is important to highlight that inflammatory markers are components of complex bidirectional regulatory mechanisms, and are involved in multiple overlapping processes, relating to infection and to the maintenance of homeostasis within the body. As such, the dynamic roles of cytokines, chemokines and transcription factors require further investigation by mechanistic studies.

Given the variations in methodology among much of the existing body of research in this field, there is a need for creating standard operating procedures to allow for better comparisons between studies and accelerate research and development within this area. In the future, it is likely that this field of research will rely on the collective efforts of researchers in nutrition, epidemiology, medicine, immunology and microbiology, requiring more interdisciplinary collaboration to enable a more complete understanding of the complex interactions between the host and gut microbiota.

## Figures and Tables

**Figure 1 ijerph-17-07618-f001:**
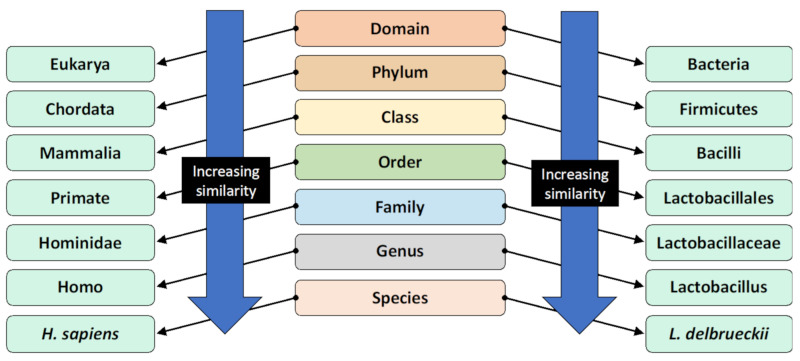
The taxonomic classification system and the classification of humans and *Lactobacillus delbrueckii* as an example [8]. Organisms are classified by this hierarchical system, where the most general and inclusive group is at the domain level. Organisms within the same species are most genetically similar.

**Figure 2 ijerph-17-07618-f002:**
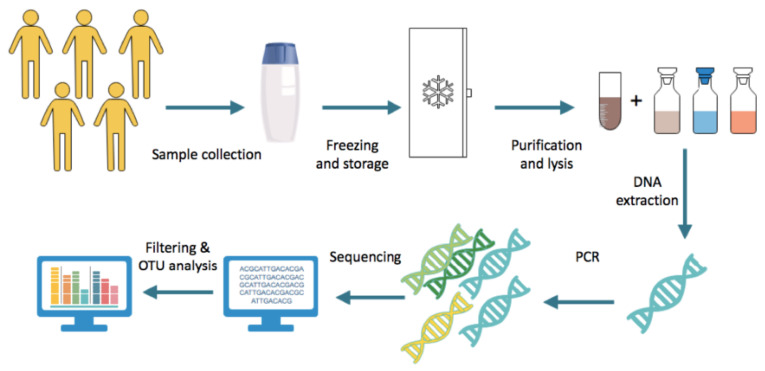
Steps generally undertaken in stool microbiome analysis.

**Figure 3 ijerph-17-07618-f003:**
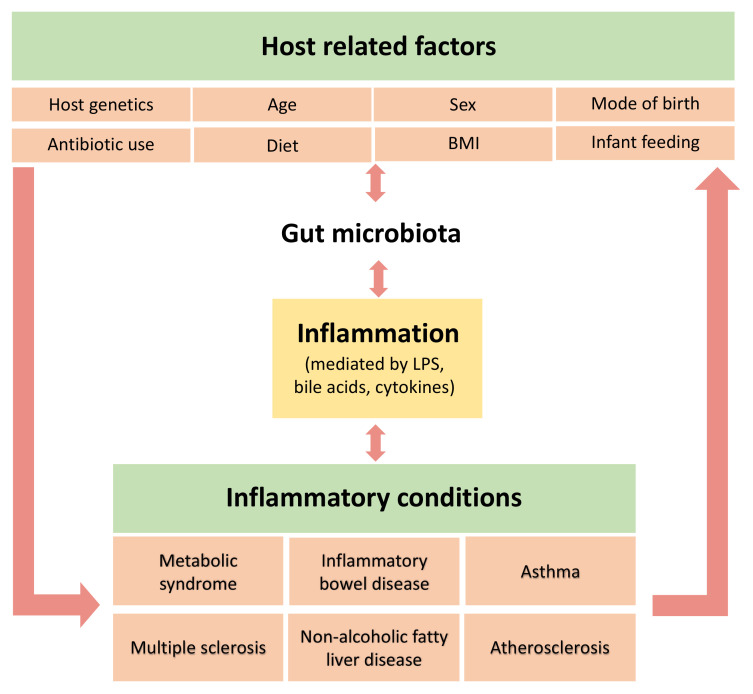
Interrelationships between the gut microbiota, inflammation and inflammatory conditions. The gut microbiota is shaped by various factors and has a bidirectional relationship with diet and BMI. It also has a bidirectional relationship with inflammation and depending on its composition, it can inhibit or stimulate inflammatory pathways. These, in turn, can promote the onset of various inflammatory conditions.

**Table 1 ijerph-17-07618-t001:** The inflammatory nature of cytokines [98,101,102].

Pro-inflammatory	Interleukin-1β
	Interleukin-8
	Interleukin-12
	Interleukin-18
	Interleukin-23
	Tumour necrosis factor-α
	Monocyte Chemoattractant Protein-1
Anti-inflammatory	Interleukin-10
	TGF-β
	Interleukin-4
	Interleukin-27
	Interleukin-35
Variable	Interferon-α *
	Interleukin-6 *

* Contrasting mechanisms demonstrate that this cytokine is both involved in pro- and anti-inflammatory processes [101,103].

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
