# Peer review of "The Gut Microbiota and Inflammation: An Overview"

_ijerph, 2020, doi:10.3390/ijerph17207618_

Round 1
Reviewer 1 Report
I think this is a well-written manuscript. After careful review, I did not find any typographical errors in the manuscript.
I believe that this review has many strengths.
It is inclusive of a variety of topics relating to inflammation and it is timely in that it covers inflammation-related disease states that are rising in incidence worldwide.
I think the review is appropriate in its breadth and depth for a review article.
The review has few weaknesses although
I would prefer to see expansion of certain areas within the paper, particularly the section on host genotype (section 5.4).
I think that inclusion of research relating to HLA and inflammatory-related illnesses like autoimmune diseases would greatly strengthen the paper.
Reviewer 2 Report
This is a well written review about the available evidence on gut microbiota and inflammation markers. Several association are discussed as well as mechanisms. The topic is of general interest for current research.
Obviously this review is not a “novelty” and maintains all the limits of a “narrative” review. Data from studies are only described without a critical analysis of the association.
The chapter on inflammatory bowel disease is poor and not essential for the aim of the review. Authors may consider to delete it.
Bibliography is quite complete.
Reviewer 3 Report
Dear Authors,
This is a very interesting manuscript looking into 'novel' and important public health issues.
Please find below several comments/suggestions:
Line 169 - 'protein' any protein? or animal protein? you compare this to a plant based diet
Line 258 - you mentioned '...and are associated with...' but it is not clear; do we observed increase of LPS?
Line 259 - 'LPS are translocated from bacteria and enter the circulation' - how is LPS translocated? human guts have antimicrobial barrier; so how LPS can be moved from the guts into the circulation? this should be explained
Line 260 - 'In turn, this leads to an increase in gut permeability due to a decreased expression of tight junction proteins in enterocytes, and promotes macrophage infiltration in adipose tissue' - this is not clear; does translocation of LPS increase gut permeability? and how does it link to adipose tissue?; is this an effect on adipose tissue inflammation only?
Line 298 - '... acetate may consequently be linked with weight gain' - is there any evidence for this?
Also, you tried to discuss many different aspects of the relationship between inflammation and the gut microbiota; as this is a very complex subject it is not really possible to discuss in depth all aspects in one review paper so it is not surprising that some parts of this manuscript are quite superficial. My recommendation is to change the title - maybe to 'The gut microbiota and inflammation - overview'.
Reviewer 4 Report
The manuscript describes about influencing of the gut microbiota in the inflammation pathway related to chronic disease. The theme is relevant in this field and interesting to the readerships. It's well written, but sometimes vague. The authors should organize better to improve that. Some point should be reviewed.
1. The number of the topic could begin in the topic 2; i.e, "Introduction"; "1. What is the gut microbiota?"...
2. The first two topics (1 and 2) are too short, so the authors could get it together.
3. Considering gnotobiotic models one kind of the animals study. The authors could discuss better other studies using animals.
4. You should get details about cell culture study.
5. Please, correct the word "microbiota" in the 147 line.
6. Considering diet have strong influence on microbiota, the authors should describe better about interacting nutrient x microbiota.
7. Explain about the LPS activating the TLR pathway promoting the inflammation associated with chronic disease. You should associate LPS with the inflammation pathway in more detail.
8. Complete that topic describing SCFAs acting in the central pathway contributing with energetic homeostasis.
9. What's the mechanism that Propionate inhibits pro-inflammatory cascade? Please, describe that.
10. Explain the cytokines related to gut micobiota. Why did you choose only TNF-alpha and IL-6 to explain?
11. You could add this topic in the LPS topic and include the LPS mechanism.
12. I suggest you provide a table with the manuscripts and the contents researched.
Round 2
Reviewer 4 Report
The authors have done the revisions suggested improving the manuscript.